

# Fragility Index, power, strength and robustness of findings in sports medicine and arthroscopic surgery: a secondary analysis of data from a study on use of the Fragility Index in sports surgery

Aleksi Reito[1,2], Lauri Raittio[3] and Olli Helminen[1,4]

[1] Department of Surgery, Central Finland Hospital, Jyväskylä, Keski-Suomi, Finland
[2] Coxa Hospital for Joint Replacement Ltd, Tampere, Pirkanmaa, Finland
[3] Medical School, University of Tampere, Tampere, Finland
[4] Cancer and Translational Medicine Research Unit, Oulu University Hospital, Oulu, Pohjois-Pohjanmaa, Finland

## ABSTRACT

**Background:** A recent study concluded that most findings reported as significant in sports medicine and arthroscopic surgery are not "robust" when evaluated with the Fragility Index (FI). A secondary analysis of data from a previous study was performed to investigate (1) the correctness of the findings, (2) the association between FI, $p$-value and post hoc power, (3) median power to detect a medium effect size, and (4) the implementation of sample size analysis in these randomized controlled trials (RCTs).

**Methods:** In addition to the 48 studies listed in the appendix accompanying the original study by *Khan et al. (2017)* we did a follow-up literature search and 18 additional studies were found. In total 66 studies were included in the analysis. We calculated post hoc power, $p$-values and confidence intervals associated with the main outcome variable. Use of a priori power analysis was recorded. The median power to detect small ($h > 0.2$), medium ($h > 0.5$), or large effect ($h > 0.8$) with a baseline proportion of events of 10% and 30% in each study included was calculated. Three simulation data sets were used to validate our findings.

**Results:** Inconsistencies were found in eight studies. A priori power analysis was missing in one-fourth of studies (16/66). The median power to detect a medium effect size with a baseline proportion of events of 10% and 30% was 42% and 43%, respectively. The FI was inherently associated with the achieved $p$-value and post hoc power.

**Discussion:** A relatively high proportion of studies had inconsistencies. The FI is a surrogate measure for $p$-value and post hoc power. Based on these studies, the median power in this field of research is suboptimal. There is an urgent need to investigate how well research claims in orthopedics hold in a replicated setting and the validity of research findings.

Corresponding author
Aleksi Reito, aleksi@reito.fi

## INTRODUCTION

Evidence-based medicine (EBM) has become the cornerstone of current healthcare systems. The aim of EBM is to yield robust, well-performed clinical research studies (*Barkun et al., 2009*). A single study should be valid, robust and therefore reproducible (*Errington et al., 2014*). A robust study indicates an adequately powered study setting with high signal-to-noise ratio (*Loken & Gelman, 2017*). During recent decades, however, increasing concern has been expressed over whether the claims of clinical research are truly unbiased and valid (*Ioannidis, 2005*; *Open Science Collaboration, 2015*; *Nosek & Errington, 2017*; *Altman, 1994*). Substantial efforts have been made to establish the factors that are associated with low replication rates and poor reproducibility, and also to investigate the severity of the problem (*Munafò et al., 2017*; *Goodman, Fanelli & Ioannidis, 2016*; *Loken & Gelman, 2017*).

One of the key elements is inadequate power to detect a statistically significant finding: the lower the power, the lower the probability of detecting a statistically significant effect under the assumption that the true effect is of a particular size. A power value of 50% indicates that out of 100 identical studies on average only 50 will produce a significant finding if a particular sized effect under investigation exists (*Button et al., 2013*; *Ioannidis, 2005*). Combining low powered studies and publication bias, that is, significant results are more likely to get published, the likelihood of successful replications are inherently poor (*Button et al., 2013*; *Szucs & Ioannidis, 2017b*; *Ioannidis, 2005*). This, most importantly, is due to the fact that if, against the odds, an underpowered study produces a nominally significant finding, it will overestimate the results meaning that the reported effect size is exaggerated compared to the population value (*Colquhoun, 2014*; *Ioannidis, 2008*). For example, with a power of 46%, the average effect size among only the significant findings will be 1.4 times higher on average than the true effect (*Colquhoun, 2014*). This is known as the inflated effect, and subsequent replication studies with higher power will result in smaller effects, possibly even contradicting the results of the initial study (*Button et al., 2013*). Terms "winners curse" and "proteus phenomenon" have been coined to describe these issues (*Button et al., 2013*).

Another major issue is the deep-rooted habit of making statistical inferences solely on a dichotomous basis (significant/not significant), using 0.05 as an arbitrary cut-off, whereas the $p$-value is a continuous measure of the compatibility of the data to the null model (*Goodman, 1999*; *Szucs & Ioannidis, 2017b*). This has resulted in the undesirable practice of pursuing nominal statistical significance and novel findings so as to attract the attention of publishers (*Ioannidis, 2005*). The most extreme example is $p$-hacking which has been shown to be present in the orthopedic literature (*Bin Abd Razak et al., 2016*).

The recent study by *Khan et al. (2017)* addressed a very important topic related to these issues. They investigated the median value of the Fragility Index (FI), a recently proposed index which denotes the minimum number of cross-over patients needed to change a significant into a non-significant finding (*Walsh et al., 2014*). This theoretical approach is associated with the strength of research claims, or fragility as defined by *Walsh et al. (2014)*. *Khan et al. (2017)* conclude that the most significant findings in sports medicine

and arthroscopic surgery are not "robust." Their claims are fully supported, since the median value of the FI was just 2, that is, changing two patients between study groups would reverse the significance. This runs contrary to our hopes, since we wish to base our clinical decisions on robust and valid studies. It is noteworthy to remark that low FI is not explicitly a bad thing since if the true effect indeed corresponds to low FI, it may be the smallest number of patients to show evidence of an effect and can be considered justified from an ethics perspective.

For studies to be robust and have good validity, they must also be replicable and reproducible. Replicability or reproducibility, which are directly associated with power, are an under-researched topics in orthopedics as in clinical research generally. Therefore, to assess the wider implications of the FI in the assessment of findings in sports medicine and arthroscopic surgery, we performed a secondary analysis of the data gathered by *Khan et al. (2017)*. Aims of our study were to investigate (1) the correctness of findings in sports medicine and arthroscopic surgery randomized controlled trials (RCTs), (2) the association between FI, *p*-value and post hoc power, (3) median power to detect a medium effect size in these RCTs, and (4) the implementation of sample size analysis.

## METHODS

We performed a secondary analysis on the data studied by *Khan et al. (2017)*. For their study, they conducted a systematic literature search to identify RCTs in orthopedic sports and arthroscopic surgery that had used a 1:1 parallel two-arm design with at least one significant dichotomous outcome reported in the abstract. *Khan et al. (2017)* included studies published between January 1, 2005 and October 30, 2015. We performed a follow-up literature search in the Medline database using the algorithm described by *Khan et al. (2017)* and identified all studies published between October 30, 2015 and July 7, 2018 meeting the inclusion criteria. The first author (AR) reviewed all 330 references identified and other authors (LR, OH) reviewed 165 each. Studies flagged suitable by two reviewers were included in the analysis and any disagreement was resolved by consensus.

### Data extraction

All the studies listed in the appendix accompanying the original study and identified in the follow-up search were retrieved. The chosen outcome variable and the corresponding proportions of events between the intervention and control groups were not given in the data extracted by *Khan et al. (2017)*. Hence for each study we identified the outcome variable chosen by *Khan et al. (2017)* on the basis of the number of events reported in the appendix. This was a straightforward procedure, since the majority of the studies reported the number of events in both study groups. *Khan et al. (2017)* reported the baseline number of participants. For the statistical analyses including *p*-value and FI and power analysis, we calculated loss to follow-up by identifying the actual number of participants included in the analysis of the chosen outcome variable. Each article was also searched for a description of an a priori power analysis or sample size calculation. The values of FI reported by *Khan et al. (2017)* were recorded and also re-calculated.

## Statistical analysis

We re-calculated the *p*-values instead of extracting them from the studies. By this measure we aimed to assess whether the comparisons were correctly calculated, that is, the correctness, and detect possible statistical inconsistencies and investigate the validity of the results. At first, two-sided Fisher's exact test was used to calculate the *p*-value for each comparison of the proportion of events between intervention groups for each study. Then, we checked which method to assess between group comparison of proportions was mentioned in the methods section. If the *p*-value exceeded 0.05, we re-calculated it using the Chi-squared test with Yate's correction. If the *p*-value still remained above 0.05 we calculated the Chi-squared test without Yate's correction. Analyses were performed by the first author (AR) and all inconsistencies were confirmed by the second author (LR). Post hoc power, corresponding to the proportions of events between two groups, was calculated using the exact unconditional power of the conditional test. More precisely, the *power.exact.test* function in the *Exact* package for R was used. This same method is used by G*Power software. We calculated the risk ratio and corresponding 95% confidence interval for each selected outcome variable. When no events were reported for either group, the risk ratio was not calculated. The FI was plotted against post hoc power, *p*-value, lower limit of the 95% confidence interval and width of the confidence interval. These plots were interpreted graphically to estimate the association between the variables of interest. We also estimated a posteriori or "after-the-fact" power for each study included in the analysis. To do this, we used the sample sizes in each study to estimate the power to detect the difference for two proportions of events between study groups. Since a dichotomous outcome always involves comparison of two proportions, we used two different baseline proportions for study group 1 in each study; 10% and 30%. The observed power was then calculated using a proportion ranging from 10% to 70% for study group 2. We also calculated the respective Cohen's h to estimate the effect size for each comparison of proportions. Comparison of study characteristics (FI, sample size) was done using the Mann-Whitney *U*-test.

## Simulation data

To validate our conclusions and check our assumptions regarding the associations between the variables of interest, we performed a data simulation. Three data sets were created each with 100 cases, or hypothetical studies, to compare the statistical significance of two proportions similar to those in our data set. Each case or hypothetical study included a two-group comparison of a dichotomous outcome. The number of units or hypothetical patients per study was the same for both groups per study and it was also the same across each data set. The number of events, that is, the proportion of outcome events was randomly generated. The first data set comprised two study groups of 30 units, or hypothetical patients, per group, the second with 60 and the third with 240 per group. The numbers of events in the two groups in each data set were selected randomly from normally distributed values using the floor of the absolute value to obtain non-negative integers. For the first data set, group 1 had a mean of four events with a standard deviation (SD) of 2 and group 2 had a mean of 12 events with a SD of 4. For the second

data set with 60 baseline patients per group, the corresponding parameters were a mean of eight events with a SD of 4 and a mean of 12 events with a SD of 8. For the data set with 240 baseline cases, the parameters were a mean of 40 events with a SD of 16 and a mean of 48 events with a SD of 16. These parameters yielded the average proportion of 50% significant differences between the proportion of events in the two hypothetical groups.

## RESULTS

In the original study, *Khan et al. (2017)* included 48 studies in their analysis. After our follow-up search we identified an additional 330 potential studies published after the study by *Khan et al. (2017)*. Of these 18 were included in our study after reviewing the full texts. The final analysis comprised 66 studies. One study had unreplicable results with both data and statistical inconsistencies. In one another study there was a data inconsistency. In addition, there was a statistical inconsistency in six studies totaling eight studies with an inconsistency. A flow chart of inconsistency assessment is shown in Fig. 1. Corresponding authors of these eight studies were contacted (Supplement 1). Six authors did not respond. All authors were contacted again in 1–2 months if they did not respond to the first contact attempt. One of the authors and one study group representative responded. In the former case, the author confirmed the use of the mentioned Wilcoxon signed rank test. In the latter case the group representative responded that they had actually used the Chi-squared test instead of the stated Fisher's exact test.

The FI was associated with the obtained *p*-value (Fig. 2A). Since the FI is a discrete variable, the association is not graphically linear. The data simulations for fixed sample sizes showed that with increasing sample size the range of *p*-values to which a single FI corresponds decreases (Figs. 2B–2D). The association between post hoc power and FI was similar (Fig. 3). Post hoc power was directly associated with the significance level obtained (Fig. 4A). A similar association was seen in each of the three data simulations (Figs. 4B–4D).

Fragility Index was associated with the lower limit of the 95% CI (Fig. 5A). This association was also apparent in the simulation data sets (Figs. 5C–5E).

The median statistical power, that is, "after-the-fact" power, to detect different proportions of events and effect sizes using the sample sizes in the 66 studies are shown in Table 1. The median power to detect a medium effect size (Cohen's $h > 0.5$) with a baseline proportion of events of 10% and 30% was 42% and 43%, respectively.

A total of 50 of the 66 studies reported the parameters used in the a priori power analysis. The median number of patients per study group was 30 (range: 12–80). When comparing the FI or sample size, we not able to detect a difference in ranks between studies with a sample size calculation and without it (both $p = 0.9$).

## DISCUSSION

### Terminology

*Walsh et al. (2014)*, who originally derived the FI, suggested that it "helps identify less robust results" indicating that robustness is associated with the *p*-value of the finding. *Khan et al. (2017)* associated also the robustness to the statistical significance of the study.
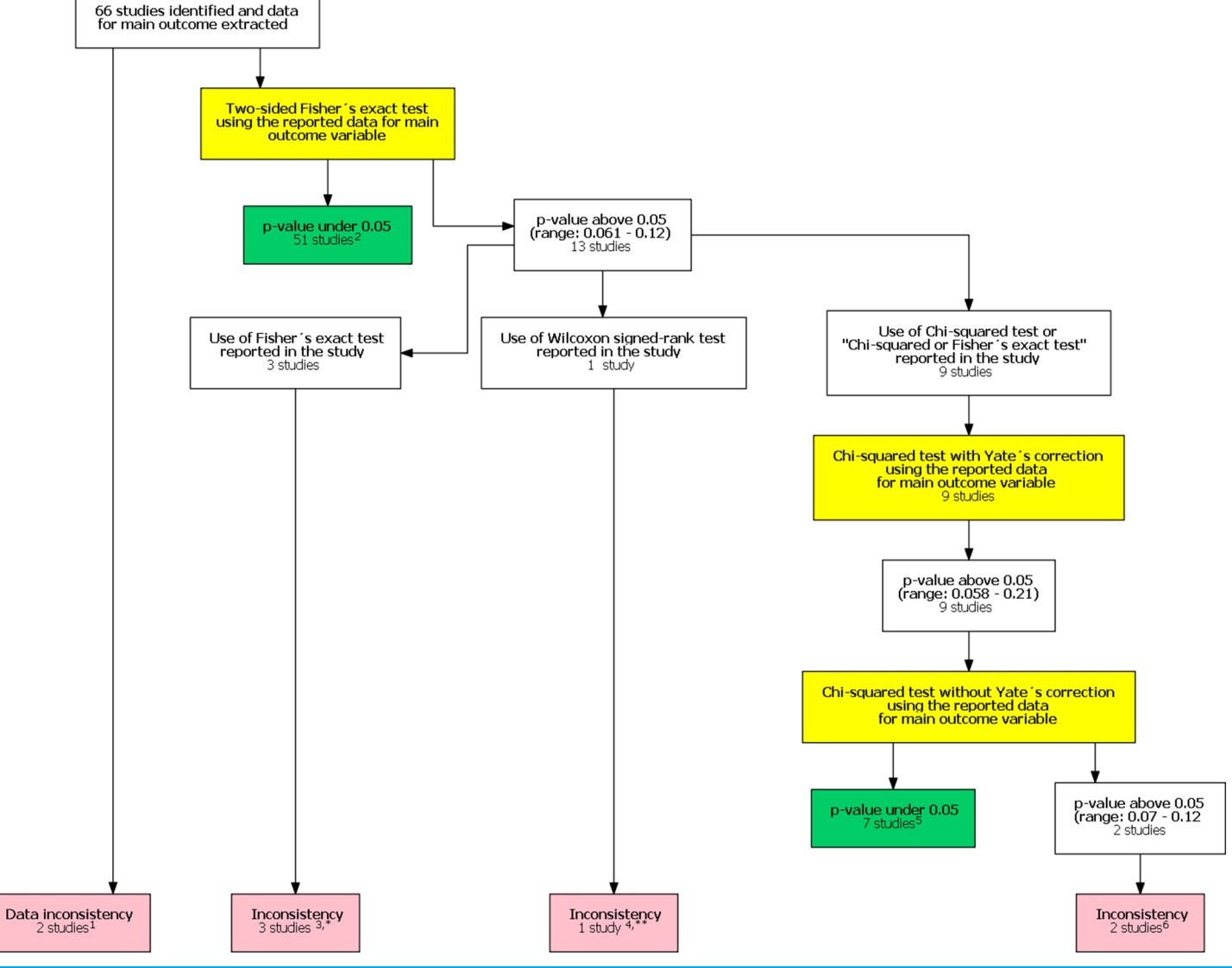

**Figure 1  Flow chart of inconsistency assessment.** Yellow boxes indicate a recalculation of the *p*-value for the main outcome variable reported in the study using the test mentioned. [1]These studies had data inconsistency. This indicated that exact number of patients with an outcome and total number of patients in each study group was not reported explicitly and these numbers could not be calculated from the stated proportions. Please see Supplement for further description. [2]For these studies, using the number of outcomes and patients reported in the study, a two sided Fisher's exact resulted to a *p*-value below 0.05 for the outcome reported in the abstract. [3]In these studies it was stated that Fisher's exact test was used. Using the number of outcomes and patients reported in the study, a two sided Fisher's exact resulted to a *p*-value above 0.05 for the outcome reported in the abstract. [4]In one study the authors reported a difference in proportions using Wilcoxon signed rank test. [5]For these studies, using the number of outcomes and patients reported in the study, a Chi-squared test without Yate's correction resulted to a *p*-value below 0.05 for the outcome reported in the abstract. [6]In two studies using the number of outcomes and patients reported in the study, Chi-squared test without Yates correction resulted to a *p*-value above 0.05 for the outcome reported in the abstract. [*]One study group representative responded and claimed that opposite to their methods, describing the use of Fisher's exact test they actually had used a Chi-squared test. [**]Authors responded and confirmed using Wilcoxon signed rank test to compare proportions.

We intend to associate robustness with the replication potential of the study or finding. We define a robust study or finding as one which is yielded from a high powered study with a high signal-to-noise ratio and therefore has high replication probability (*Loken & Gelman, 2017*). The *p*-value as well as the FI have large sample-to-sample

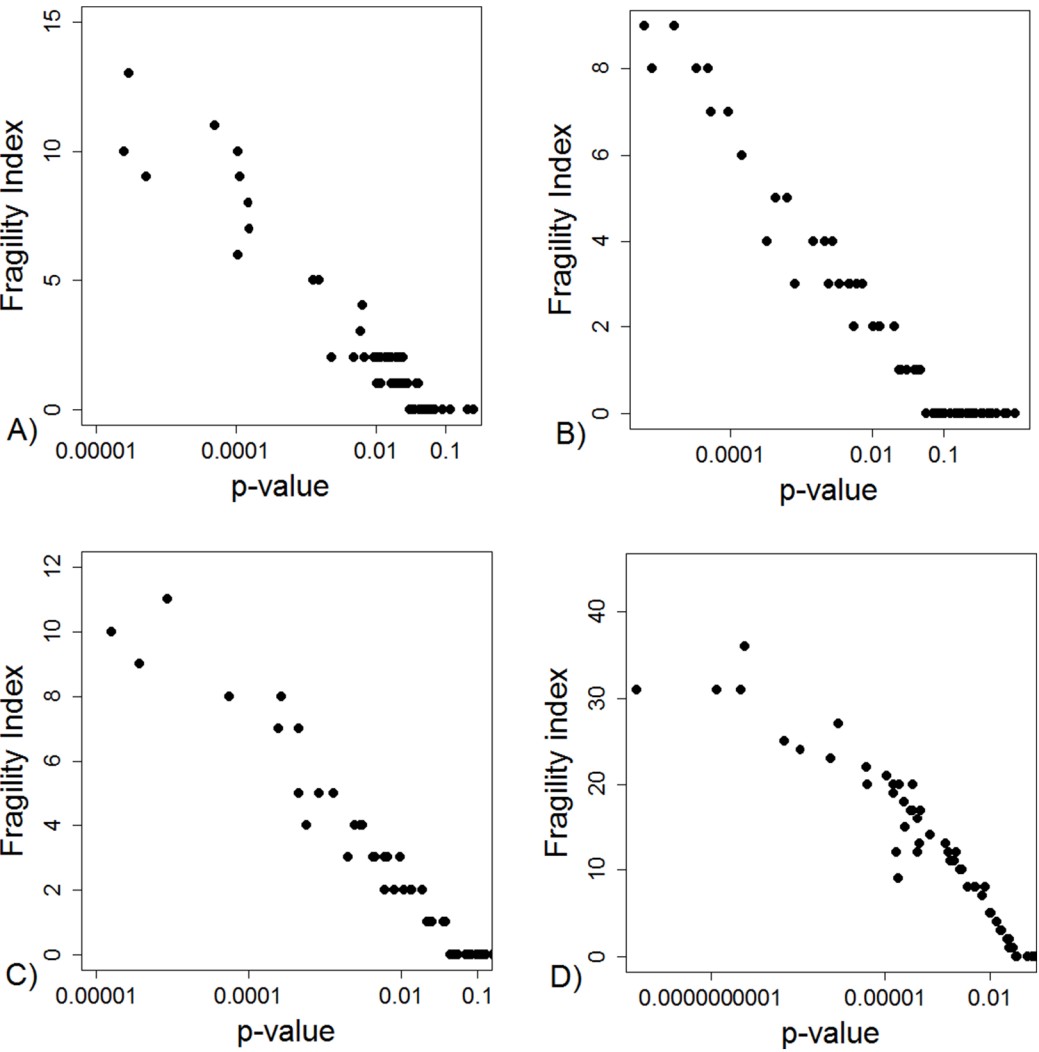

**Figure 2 Association between fragility index and *p*-value.** (A) Data from 66 studies. Simulation data sets of 100 hypothetical studies including 30 (B), 60 (C), and 240 (D) units or hypothetical patients per group in a two-group comparison of dichotomous outcome.

variation especially if obtained from low-powered studies and therefore do not contribute per se to the robustness of the finding (*Halsey et al., 2015*). This is the reason we prefer quotation marks around the term "robust" when referring to works by *Walsh et al. (2014)* and *Khan et al. (2017)*. Only when extremely low *p*-values (or high FIs) are concerned, can we expect a high probability of successful replication and in these instances the *p*-value contributes to the robustness. Similar to *absence of evidence is not evidence of absence*, a finding associated with a high or "non-significant" *p*-value cannot be considered automatically as robust (or fragile) unless power, uncertainty and signal-to-noise is thoroughly assessed. Finally, study validity indicates the degree of conclusions can be taken from the study and validity also means that appropriate test statistics have been used.
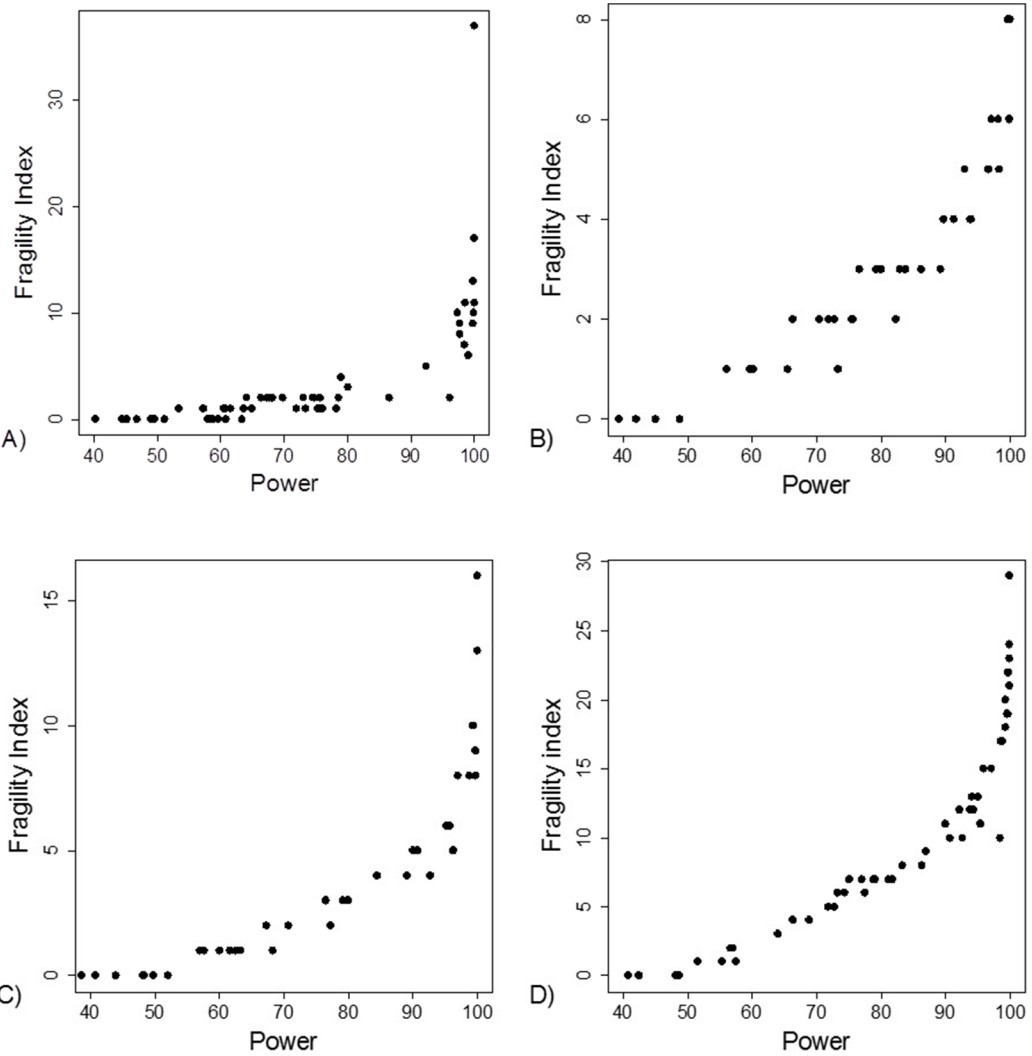

**Figure 3 Association between post hoc power and fragility index.** (A) Data from 66 studies. Simulation data sets of 100 hypothetical studies including 30 (B), 60 (C), and 240 (D) units or hypothetical patients per group in a two-group comparison of dichotomous outcome.

## Key findings

A thorough investigation of the outcome variables reported in 66 RCTs in sports surgery revealed data or statistical inconsistencies in eight (12%) studies. We view this as a concern. The inconsistencies varied from statistical inconsistencies to data issues. By the former we meant unreplicable results or inconsistent use of statistical tests, whereas by data issues we meant data reporting was not explicit, only the percentage of patients with the event of interest was reported, whereas the actual number of events was not disclosed. It was not possible to assess whether the inconsistencies were due to actual calculation error or a falsely reported event prevalence. In these cases, readers are left with the task of evaluating the robustness, subjective meaning and applicability of a study unsure of whether the basic statistical measurements are correct.
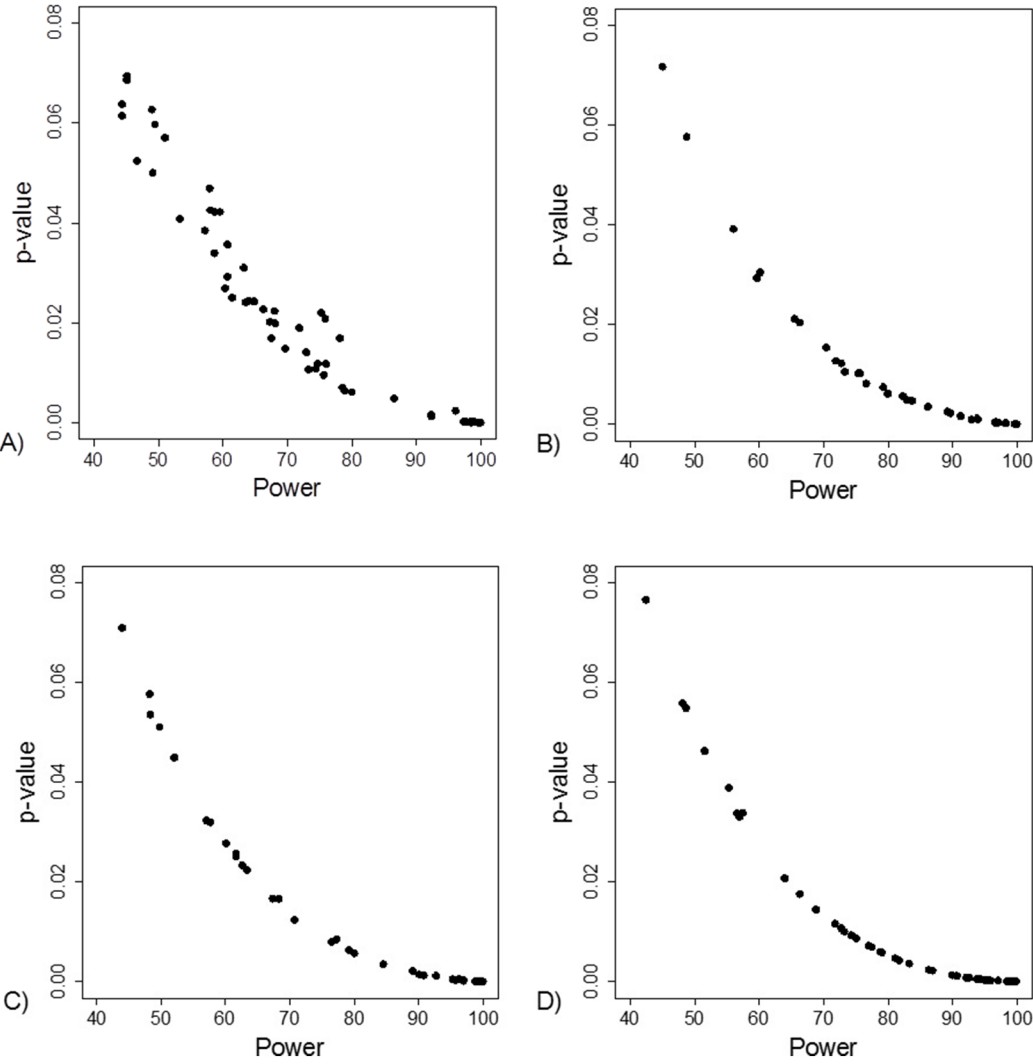

**Figure 4** **Association between post hoc power and *p*-value.** (A) Data from 66 studies. Simulation data sets of 100 hypothetical studies including 30 (B), 60 (C), and 240 (D) units or hypothetical patients per group in a two-group comparison of dichotomous outcome.

Another concern is related to the poor statistical power. The median power to detect a medium effect size with a baseline proportion of events of 10% and 30% was 42% and 43%, respectively, among the 66 RCTs included in the review. It is clear that were we to run identical trials for each study included in both this review and in that of *Khan et al. (2017)*, we would end up with several studies with conflicting results compared to those of the original trials owing to the poor statistical power. Poor statistical power, that is, a low probability of detecting effect sizes that would be of clinical or other importance if they exist, is directly associated with small sample size. It is of upmost importance to note that the problem of low power in our field is not new since it was highlighted already a decade ago by *Freedman, Back & Bernstein (2001)* and more recently by *Abdullah et al. (2015)*.

This scenario, namely poor reproducibility and low replication success, has manifested itself in many other research disciplines, most importantly cancer research and psychology,

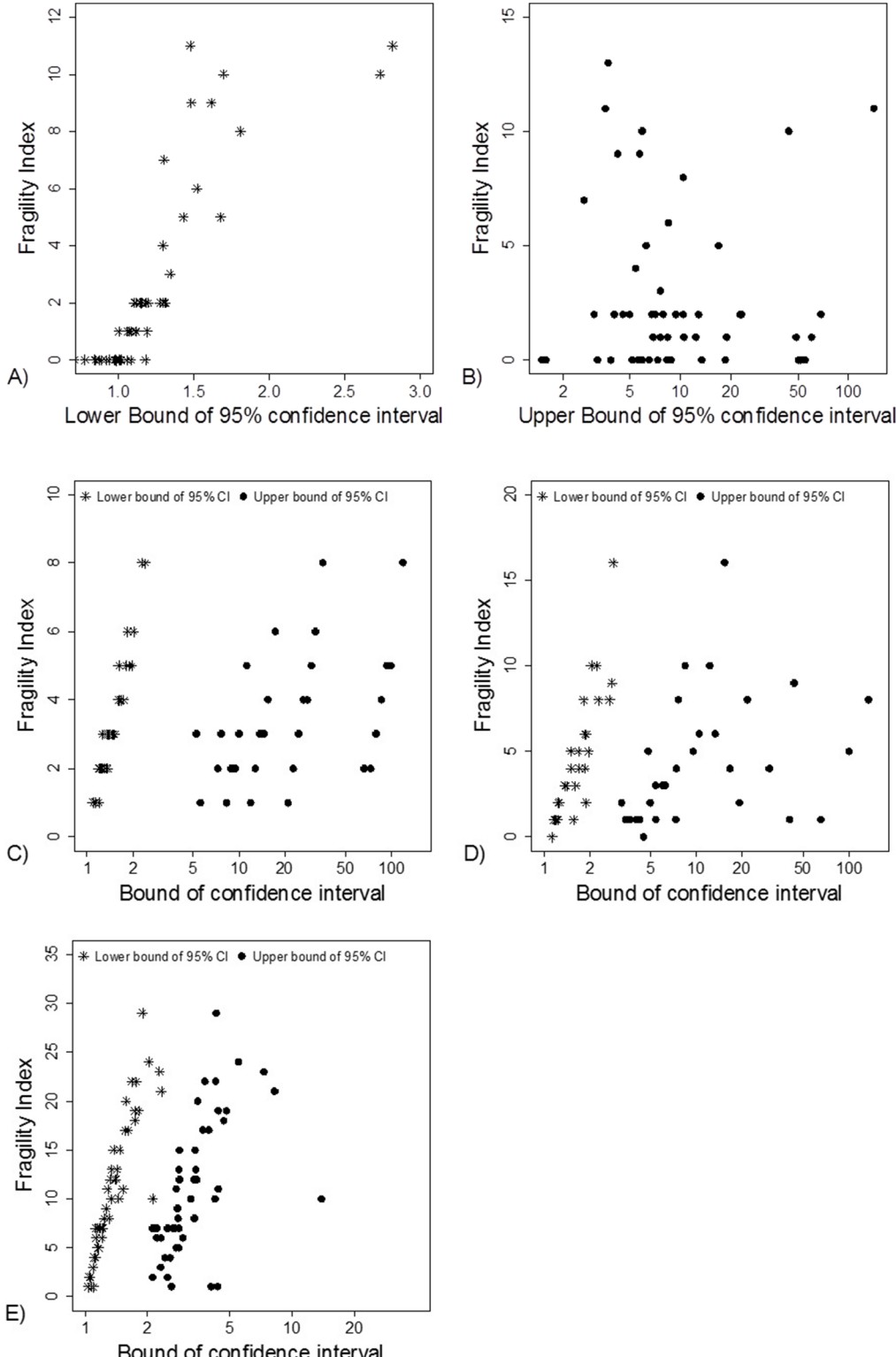

**Figure 5 Association between FI and limits of confidence interval.** (A) and (B) Data from 66 studies. Simulation data sets of 100 hypothetical studies including 30 (C), 60 (D), and 240 (E) units or hypothetical patients per group in a two-group comparison of dichotomous outcome.

**Table 1 Median "after-the-fact" power to detect defined differences in the proportions of events between two study groups and their corresponding effect sizes using Cohen's h.**

| Proportion of events in Group 2 | Cohen's h | Median power in the 66 studies included |
|---|---|---|
| Baseline proportion of events in Group 1 = 10% | | |
| 20% | 0.28 (small effect) | 14% |
| 29.3% | 0.50 (medium effect) | 42% |
| 30% | 0.52 (medium effect) | 44% |
| 40% | 0.73 (medium effect) | 75% |
| 50% | 0.93 (large effect) | 93% |
| Baseline proportion of events in Group 1 = 30% | | |
| 10% | 0.52 (medium effect) | 44% |
| 20% | 0.23 (small effect) | 11% |
| 30% | 0 | 3% |
| 40% | 0.21 (small effect) | 10% |
| 50% | 0.41 (small effect) | 30% |
| 54.4% | 0.50 (medium effect) | 43% |
| 60% | 0.61 (medium effect) | 61% |
| 70% | 0.82 (large effect) | 87% |

**Note:**
A median power of 44% to detect an effect size of 0.52 with a baseline proportion of events of 10% means that retrospective power analysis was performed for each of the 66 studies on the assumption that 10% of the patients in arm 1 and 30% in arm 2 have the target event. The median power observed among the 66 studies was calculated and reported as shown above. The same assessment was done with a varying proportion of events in group 2 and with another baseline proportion of events (30%). A median power of 40% indicates a 40%, chance of obtaining a significant result with the sample sizes used in the 66 studies, assuming a true effect size of 0.52 (proportion of events 30% vs 10%).

of which the latter is considered to be facing a "replication crisis." In addition, recent reviews have shown that the average statistical power to detect an effect of particular size in neuroscience research is very low (*Button et al., 2013*; *Szucs & Ioannidis, 2017a*). A suggested reason is an increase in research flexibility, which allows the search for smaller effects, while sample sizes have remained unchanged (*Button et al., 2013*). This has led to a substantial decrease in statistical power. As the authors of that review state, researchers nowadays aim for smaller and more subtle effects than earlier, when "low-hanging fruit were targeted" (*Button et al., 2013*). Whether we can rule out a similar situation, that is, poor replication and reproducibility, in the orthopedic literature is by no means clear.

Post hoc power, that is, the probability of detecting the effect size seen in the study, and *p*-value were directly associated with the FI in our secondary analysis. Hence the FI is a surrogate measure for post hoc power (*Carter, McKie & Storlie, 2017*). Thus, when the FI value is small, the post hoc power will be close to 50% and there should be doubts regarding the strength of the finding meaning that the finding in question cannot be considered automatically robust. The infamous *p*-value cut-off of 0.05 corresponds to a post hoc power of 50% (*O'Keefe, 2007*; *Hoenig & Heysey, 2001*). All non-significant results (*p* > 0.05) show a post hoc power of less than 50% (*Hoenig & Heysey, 2001*). This is also evident in our analysis. If an experiment results in a one-tailed *p*-value of 0.05, there is an 80% chance that a replication study will show a one-tailed *p*-value between 0.00008 and 0.44 (*Cumming, 2008*). In their original study, *Khan et al. (2017)* reported a strong

linear correlation between the p-value and FI. This is only reasonable, since the FI is inherently associated with the strength of the research findings included in its assessment, as shown in our study. The further the confidence interval limit departs from the "null," the stronger the study effect. This is rather like comparing two normally distributed variables and their confidence intervals. Finally it is of importance to note that neither the p-value nor the FI can give definitive evidence as to whether the finding is indeed true. For this matter we should estimate the prior probability that our effect truly exists in the population under investigation (*Ioannidis, 2005*). To be more specific, even if the prior probability of the effect (Cohen's $d = 1$), being true is 0.5, which could be considered exceedingly high, for a p-value of 0.0475 with a sample size of 32 (16 + 16), the false positive rate (FPR) is as high as 26% (*Colquhoun, 2014*, *2017*). In other words one out of four studies would produce a false positive finding. To the best of our knowledge, the prior probability of an effect and a FPR associated with it has never been implemented in the field of orthopedics.

To sum up, the FI, power, p-value and magnitude of effect are, by definition, all associated. Hence the FI per se does not contribute a new parameter for evaluating a research finding. Neither does it help to estimate the FPR. Furthermore, it does not rule out the fact that p-values show great sample-to-sample variability, thus poorly reflecting the true effect under investigation and predicting future replication studies in the case of excessively small sample sizes. However, we fully agree with *Khan et al. (2017)* that it is worthwhile investigating how reporting the FI would improve clinician's ability to detect trials that should be viewed cautiously. Misconceptions and misinterpretation of the statistical parameters, namely p-value, confidence intervals and power is common and therefore, FI may provide a clinically practical parameter (i.e., number of individual patients) to evaluate the strength of a research finding compared to these other parameters.

## Implication for future studies

As discussed above, the FI accompanies power, p-value and effect size, all of which are inherently associated with each other. All these parameters thus concern the strength of a single research finding. However, robustness is the key factor when considering the implications of research claims based on single studies. In order to evaluate the robustness of a finding we must assess the statistical power and the signal-to-noise ratio. Robust findings, that is, findings obtained from well-powered studies with high signal-to-noise ratio, are more likely to be replicated, which is a fundamental scientific principle. However, as researchers, we should expand our understanding beyond merely the robustness of our studies.

The increasing concern about the research replicability should prompt us to evaluate the current state of replicability and reproducibility in our own discipline (*Button et al., 2013*; *Nosek & Errington, 2017*; *Errington et al., 2014*). We propose that this could be assessed by a replication/reproduction triad, as shown in Fig. 6.

Above, we have discussed various means of evaluating the strength of research findings, for which the FI provides an ingenious method when evaluating 1:1 parallel-arm RCTs with a dichotomous outcome. Even if we have achieved a robust research finding,

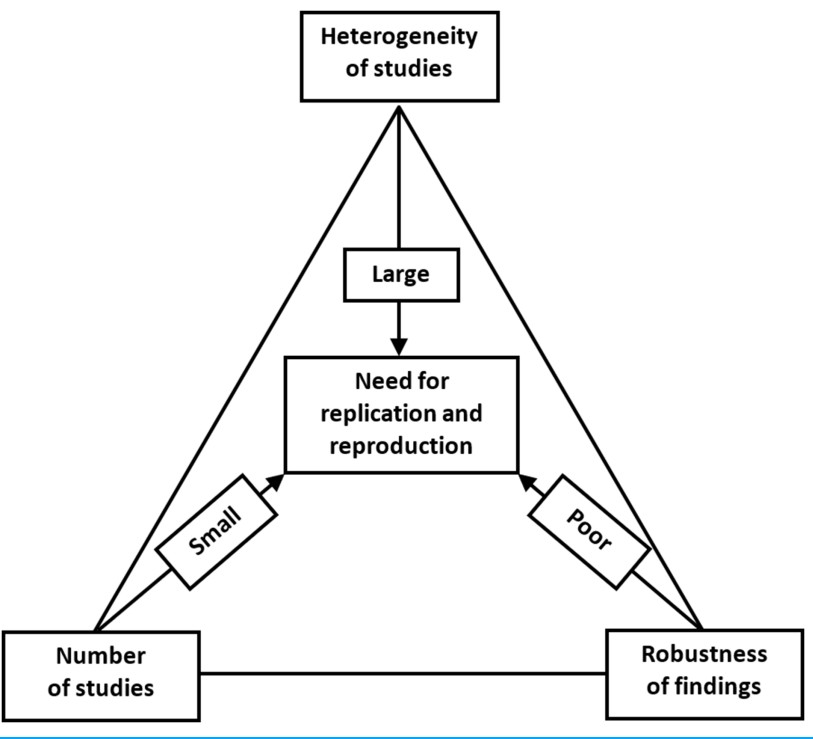

**Figure 6 Replication/reproducibility triad.**

a single study remains just a single study. We can be more confident about the research claim and apply it in clinical decision making if we can find a repeat of the same phenomenon that points toward the true population effect. Hence, it is important to be cognizant of the number of studies that have already investigated the target effect when evaluating the need for replication or reproducibility. Moreover, even when a single well-conducted study yields highly significant and/or robust findings, a replication study will still be needed. This could be done in a different geographical location by different researchers with different personal inclinations. Varying the parameters will aid in the generalizability of the effect, if comparable to that of the original study.

In addition to robustness and the number of existing similar studies, the possible heterogeneity of such studies is the third important reason for replication. Even if we have a sufficient number of studies, and all with robust estimates of effect size, we might nevertheless observe a high level of heterogeneity between them. Within-study heterogeneity is always to some degree present since it is the consequence of random variability or sampling error, since the samples used in a single study will not be identical. Between-study heterogeneity is caused by factors associated with study characteristics such as design, participant recruitment, variations in treatment, etc. Two studies may yield different estimates of effect size and, if no differences in study characteristics are found, then we will observe high heterogeneity. This variability in effect size may be due to random variation in the effect size estimate due to small sample size. Or if, for example, patients who were clearly older were sampled in one study and younger patients in the other, the observed heterogeneity can be attributed to this variation in participant

characteristics and a further replication study would be recommended to include both patient subgroups. Heterogeneity is most prominently featured in meta-analyses. Although the different methods of heterogeneity quantification in meta-analyses have their limitations, heterogeneity should not be overlooked since it is an important contributor to the robustness and reproducibility of study findings and claims.

## CONCLUSIONS

*Khan et al. (2017)* have continued the debate in an important issue when questioning the robustness of research findings in the field of orthopedic sports and arthroscopic surgery. Linked with the poor median value of the FI, the median statistical power of the studies in this field are far from optimal. This is a direct consequence of small sample sizes being utilized. Although not directly generalizable to other surgical and orthopaedic subfields, it is clear that further investigation and assessment of the validity, robustness and replicability of clinical findings is needed. Although replication and reproduction are integral components of scientific work, it remains marginal at best, overlooked in the search for "novel" findings and new discoveries. We suggest that future work should highlight the aspects related to the replication/reproduction triad introduced in this article. In sum, there is an urgent need to investigate how well research claims in orthopedics hold in replicated settings and the true validity of research claims in the field.

### Funding
The authors received no funding for this work.

### Competing Interests
The authors declare that they have no competing interests.

### Author Contributions
- Aleksi Reito conceived and designed the experiments, performed the experiments, analyzed the data, contributed reagents/materials/analysis tools, prepared figures and/or tables, authored or reviewed drafts of the paper, approved the final draft.
- Lauri Raittio conceived and designed the experiments, performed the experiments, analyzed the data, contributed reagents/materials/analysis tools, authored or reviewed drafts of the paper, approved the final draft.
- Olli Helminen conceived and designed the experiments, performed the experiments, contributed reagents/materials/analysis tools, authored or reviewed drafts of the paper, approved the final draft.

### Data Availability
    The raw data is available at OSF (https://osf.io/j7y53/).

# PeerJ

## Supplemental Information

Supplemental information for this article can be found online at http://dx.doi.org/10.7717/peerj.6813#supplemental-information.

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
