# Peer review of "Fragility Index, power, strength and robustness of findings in sports medicine and arthroscopic surgery: a secondary analysis of data from a study on use of the Fragility Index in sports surgery"

_PeerJ, doi:10.7717/peerj.6813_

## Round 0.1 · original submission · Major Revisions

The reviewers have made what I think should be useful comments when revising your manuscript. Both reviewers are supportive, as am I, of research in this area and clarity is vital for a topic with such general interest. Related to this, the online data URL did not point to a specific file and I think including the actual code used for the analyses is necessary to ensure complete transparency around your own methods.

·

Basic reporting

As I explain below, argumentation and referencing seem relatively poor to me.

Experimental design

Hard to judge, because so many things remain unclear about the methods.

Validity of the findings

Hard to judge, because so many things remain unclear about the methods and the aim of the study.

Additional comments

I appreciate the general aim and the efforts by the authors of this manuscript. After reading the manuscript, however, I have mixed feelings, partly because there are so many things that I do not understand or that remain unclear. I found many sentences about general issues related to the replication crisis that seem somewhat out of place, sometimes even without a reference. For example, in Line 222, it is said that misconceptions about "these other parameters" in statistics are common. Yes, and why do you say this, at the end of the paper? What are "these other parameters"? In Line 228, it is said that "Robust findings are more likely to be replicated, which is a fundamental scientific principle." Unfortunately, it remains unclear what the authors mean with "robust".

It seems that the authors define a finding as "robust" if chances that a replication is significant are (hopefully) high (e.g., Line 78: "Replicability, which is directly associated with power,..."). However, the presence or absence of significance is a very poor indicator of "robustness" and replicability (see, e.g., Goodman et al. 2016, What does research reproducibility mean? Science Translational Medicine 8, 341, and Amrhein et al. 2017, The earth is flat (p > 0.05): significance thresholds and the crisis of unreplicable research. PeerJ 5, e3544).

So it seems to me that while the authors are highly skeptical about significance testing (e.g., "The infamous p-value cut-off of 0.05", line 202), they fall themselves trap to overemphasis of statistical significance as the main indicator of robustness. This greatly contributes to my general feeling that the manuscript needs a lot of streamlining, a better focus, and better explanations.

L53. I would delete "even", because it may lead to the misinterpretation that power could also apply if the effect does not exist.

L54. I don't understand this sentence.

L59. There are several names for the phenomenon, and several references could be cited. "This is known as the inflated effect" sounds oversimplified and unclear to me.

L64. I would delete "always". P should be taken as a continuous measure of compatibility between the data and the model, yes, but this is often not the case. Some references should be cited here, e.g. to the recent papers by Sander Greenland.

L68. I don't understand why p-hacking should lead to a dearth of replication studies.

L104. Unclear how recalculating p-values can be used to assess robustness of a finding. To judge robustness of an effect, replications would be needed.

L108. Please explain why you did this. The description sounds as if someone wants to push p below 0.05 by all means.

L115. Explain where the FI was taken from.

L144. Reword the sentence. "Both were seen on one" is not understandable.

L147. What do you mean with "significance was higher"? The entire procedure used in this study is rather unclear to me. Why was all this re-testing done? Were the original p-values all <0.05, and you showed that you found p>0.05 when using a different test? I can hardly see any sense in that so far. The sizes of p-values often change when different tests are used, but this does not argue against robustness of results, it argues against making strong decisions based on p-values or other statistics.

L151. Again, please explain where you took the FI from. Did you calculate it yourself?

L172. Again, please explain why you think that different p-values, obtained from re-analysis of data using different tests, is a "statistical inconsistency". Perhaps I missed something?

L203. "All non-significant results show a post-hoc power of less than 50%". I see no sense in making such a statement. Did you find this relation yourself? If not, give a reference. It has long been shown that observed power is a 1:1 function of the p-value (see, e.g., Hoenig & Heisey 2001, The abuse of power: The pervasive fallacy of power calculations for data analysis. The American Statistician 55, 19-24). Observing low power in non-significant studies shouldn't be a cause for concern.

·

Basic reporting

Clear and accurate (but see comments)

Experimental design

n/a it is a theoretical study

Validity of the findings

OK

Additional comments

This paper provides a bit more information than Khan et al, and should be of interest to people in the field. The frequency of statistical inconsistencies is alarming and it's important that they are published.

The fragility index is an ingenious idea. It would certainly cast doubt on the strength of conclusions if only one or two people had to me moved into the non-responder category to change statistical significance.

The problem is that it is already known (and is shown again here) that the fragility index is strongly correlated with the p-value. To that extent it is just a surrogate for the p-value and suffers from all the problems of p-values. These are pointed out in my 2014 paper (among many other papers), to which the authors kindly refer. However they refer to it only in respect if the inflation effect. A more up to date version is at http://rsos.royalsocietypublishing.org/content/4/12/171085 and its associated web calculator: http://fpr-calc.ucl.ac.uk/
I suggest that might be a more relevant reference. I would take the view that what really matters is the false positive risk -the probability that your result is a false positive.

I'm not suggesting that the paper should be rewritten in terms of false positive risks, but it might be worth noting that the fragility index still uses the much-maligned p = 0.05 threshold. Since p = 0.05 corresponds to a false positive risk of at least 26%, the fragility index must exaggerate the reliability of the conclusions If this is mentioned it will add more strength to the conclusions of this paper concerning the lack of robustness of findings in sports medicine and arthroscopic surgery RCTs.

---

## Round 0.2 · Major Revisions

Thank you to the two new reviewers of this revised manuscript (the previous reviewers were not available to re-review). They were, like the original reviewers and myself, generally positive about the topic and interested in your treatment of it.

The question of the replicability of your analyses (a point well made by Reviewer #4) is crucial and, given the nature of your study, I’m sure you will agree here that the reader ought to be able to confirm this rather surprising/worrying finding themselves (and the supplement already helps here for those three studies). I think that Reviewer #4’s suggestion of contacting the authors where results were unable to be replicated is prudent, a point also made by Reviewer #3, as there are several possible mechanisms that could lead to this (including typos in the descriptive data, incorrect explanations of the statistical methods, and misreporting of the p-value) and it is possible that errata are required.

My best guess with Rafols, et al is that the data represents 11/30 in group 1 (as you note, this gives the reported 36.7%) but I’m not convinced by your guess of 6/27 in group 2 (giving 22.2% which is not a single digit typo away from their reported 21.1%) but rather I think more likely might be 3/27 in group 2 giving 11.1%, a single keystroke typo away from their reported percentage and now giving a Chi-squared p-value of 0.025 (not triggering Fisher’s Exact test using the rule of 80% or more of cells with expected counts of 5 or above, which would give p=0.033) and so consistent with their reported “p<0.05” in the text. The error with the percentage is then replicated in the following column. However, the same percentage (21.1%) is also included in the abstract with a p-value of 0.013, which could be a single keystroke typo away from the (two-sided) FET p-value of 0.033 for the above comparison or, much more likely I think, could be from an inappropriate application of Wilcoxon’s rank sum test on the binary outcome (one-sided p=0.013, which would match the reported p-value and require only one propagated typo in the percentage). Of course, even if I am correct, this doesn’t excuse the authors, and perhaps also the journal, for not catching this reporting error or the authors, and in this case definitely also the journal, for not reporting the data more clearly or checking their statistical methods for this analysis (they only talk about t-tests and Wilcoxon rank sum tests in their statistical methods which is partly why I favour the explanation involving the latter of these being used for the binary data). This was the first and so far only study I tried to replicate, being the first listed in your appendix, so I do not mean to suggest that there are other plausible explanations for the inability to replicate the p-values from the other studies here or the converse, but it does suggest to me that we (both the authors and PeerJ) need to be extremely confident about statements concerning these studies and I think that contacting the authors is the right and best way to achieve this. If my explanation is correct, I suspect that the authors on Rafols, et al. would be embarrassed about the errors but also annoyed if they weren’t given a chance to respond before their study was described as flawed.

As Reviewer #4 also notes, if you are able to refresh the search for new literature from the past two years, that would also strengthen the impact of this particular work. You might also be interested in looking at time trends in FI, is it getting better or worse? Similarly, comparisons could be made by journal IF if you felt that was of interest.

Some other (mostly minor) comments from me are:

Line 28 or 33: n=47 should be made clear at one of these points, but I’m not convinced that Rafols, et al. should be excluded without further investigation, including contacting the author(s).
Line 31: The use of Cohen’s effect sizes here is arbitrary and should be explicit at this point (despite it being explained on Lines 34–35). Perhaps move the latter clarification up to here? Note that Cohen himself suggested avoiding these effect sizes unless there was no better option from theory or experience and the justification later needs to be strengthened on this point.
Line 33: Give the %s (for 8 and 17) here.
Line 37: “A relatively”
Lines 53-54: A prior power, I would argue that this is the only useful kind of power calculation, is always conditional on an effect and so this text needs to include something along the lines of “under the assumption that the true effect is of a particular size”. It is nonsensical to speak of a study’s power as an absolute without this qualification (any non-trivial study has >90% power for some small effect size and also <10% power for another larger effect size). See also Lines 213–214.
Line 56: Similarly, this is not “the effect” but “a particular sized effect” and should make it clear that this would be <50% if the true effect is smaller and >50% if the true effect is larger.
Line 57: “i.e.”
Line 57: “likelihood” or “chance”
Line 70: I’d describe a p-value as a measure of the fit of the data to the null model rather than the model (which sounds more like the proposed model), but better would be as a measure of the compatibility of the data with the null model (how often would as or more extreme results be observed were there no effect).
Line 81: I’d describe what 2 means in this case to help the reader. Note that an FI just above zero is not necessarily a bad thing from an ethics perspective as it means, if the estimated effect is equal to the true effect, that the smallest number of patients necessary to show evidence of an effect were exposed to potential harms. A large FI would suggest either than the true effect is much larger than the MCID, that the study was over-powered in the first place (raising ethics concerns for RCTs), or that chance intervened. If you agree, I wonder if you could incorporate a mention of potential ethics aspects around here?
Lines 95–96: I think it is worth bearing in mind that due to these inclusion criteria, the opposite result to the n=8 “no longer significant under reanalysis” cannot be detected. This is not to say that errors with either direction of effect are acceptable, but if they occurred in both directions, I’d suggest inadequate involvement of statisticians as a possible factor; if they occurred only in one direction, I’d be much more concerned.
Lines 112–118: Were no other methods used by the authors of these studies (e.g. logistic regression or Poisson regression with robust standard errors, either without or with adjustment for explanatory variables or stratification variables, or perhaps mixed logistic regression with random centre effects for multi-centre trials, or analyses following multiple imputation)?
Line 118: Double period.
Line 156: “Chi” as per Line 155. Also Line 158 and possibly elsewhere.
Line 159: “a p-value > 0.05.” but here and elsewhere I think actual p-values would be more useful to the reader (p=0.051 might suggest a small error, p=0.510 suggests something much more important to me.)
Line 185: “finding as one”?
Line 187: “P-values” to start the sentence (and then “FIs”) but “The p-value as well as the FI” would sound more natural to me.
Line 191: “can we”
Line 192: “the p-value”
Line 195–197: This sentence is unclear.
Line 226: Check quotation as “fruit” is already plural.
Lines 234–235: This isn’t so much evident in your analysis, rather it is mathematically true. I’d suggest deleting this sentence.
Line 238: Missing space before parenthesis.
Line 239–240: If by effect you mean the difference in proportions, this is true for a fixed precision only (i.e. assuming the same sample size and standard deviation, the latter of which depends on the proportion here). Same point on Lines 242–243 and 255. Note that the strength of an effect, to me, would be the difference between the groups; this is different to the strength of the evidence of an effect or the precision of the estimated effect. If you mean standardised effect sizes, this should be made clear.
Lines 301 onwards: It’s perhaps a little surprising that you haven’t explicitly mentioned meta-analyses in your manuscript, especially given your touching on measures of heterogeneity often used there. This is up to you, but it seems to me that meta-analyses, and systematic reviews, are highly connected with questions about the robustness and replication of study findings.
Figures 1 and 2: Log-scales on the y-axes might help the reader here.

Reviewer 3 ·

Basic reporting

Interesting concept, but the article does not feel that organized. Interestingly, it also feels somewhat post hoc driven, which is ironic given the authors main message.

Experimental design

Adequate. There has been some similar work looking at the FI and its relationship with p-values (e.g., https://academic.oup.com/eurheartj/article/38/5/346/2422087).

Validity of the findings

The review of the literature and validation of the previously published studies was not reviewed carefully. It is quite interesting and a case study with review with the authors on what might have happened could also be quite telling. Hopefully the authors have been notified so that an erratum can be published.

·

Basic reporting

There needs to be some additional English language review.
The pivotal paper by Khan is not adequately referenced.

Experimental design

See comments below

Validity of the findings

See comments below

Additional comments

Thank you for asking me to provide a post-revision review of this interesting paper looking at the Fragility Index and other measures of study strength in sports medicine and arthroscopic surgery. As far as response to the initial comments of reviewers the authors have attended well to the details and give good explanation of the changes they have made. However, I have further comments on the revised paper to make. The study as presented here is a rework of data previously published by Khan et al in 2016 (Au note: the paper does not have the correct reference for this article), and thus the major fidning of this current paper is not in the fragility of the previous 48 studies but in a) further analysis of the ‘strength’ of the studies, b) an analysis to investigate the relationship of the fragility of the studies with other factors, c) the new statistical analysis of the 48 studies.
Considering a), The results are interesting, tend to confirm previous findings, after all fragility index is not ‘new’ it is just a different way of emphasising the power problems that affect small sample sizes. Why therefore have the authors added a level of confusion by starting their discussion with a redefinition of ‘robust’, this distracts from the study and is confusing. It also makes a discussion which is already overlylong longer still.
Considering b), I understand the reason that the authors use the studies previously analysed by Khan, but that paper is 2 years old and there may be more. The authors should have taken the opportunity to add more recent papers (or state that there aren’t any) found by an appropriate search methodology to the current analysis. In my copy of the study spreadsheet in the supplemental material I am only able to see the study reference when opened in excel. We need more data on these studies, particularly considering point c below.
Considering c). This alone is a worthy of publication, the authors are stating that their analysis shows that almost 20% of papers in this literature claim significant findings which do not stand up to simple statistical reanalysis. This is a major finding and the authors are right to ‘view this as a serious concern’. Firstly this finding could be very serious for the authors of the studies, their institutions, the wider credibility of sports medicine and orthopaedic surgery, not mention PeerJ itself and its editors, as well as the authors. The Editors need to be absolutely confident in this data prior to publication, I strongly suggest a second independent statistical analysis at the minimum. It is a potentially important finding and should not be suppressed, but the reader also needs to be fully assured that it is itself correct and able to be replicated . I myself have not individually checked the original data, but it did occur to me that it might also be wise to contact the original senior or first authors to ask for data clarification. It is also vital that the reader can make their own judgement as well, an excel supplement is in my view inadequate; we should be able to see straightaway while reading the paper, and suitable data in tabular form, to allow us to understand what is happening here. At the minimum we need to see the numbers used in the analysis, drop outs, statistical tests used, p values (etc) quoted, and the results of the new analysis. As I have noted above the file I downloaded did not contain this data. Lastly on this point, the authors should be more factual in their opening paragraph of the results, the words like ‘claimed’, ‘disclosed’ or ‘..one study even..’ are better left to others, here just the facts should be reported the reader will add the rest.
General points: The paper needs English language editing

---

## Round 0.3 · Minor Revisions

As noted by the reviewer, your rebuttal didn’t include point by point responses to their questions (or mine), i.e. something seems to have gone wrong with your rebuttal letter. In any case, we have based our comments on the latest version of the manuscript and I apologise in advance if I am asking something that was intended to be answered in your rebuttal itself, but these questions are likely to be asked by readers and so should be evident in the manuscript itself in any case.

I will ask you to address the very useful and insightful comments from the reviewer along with my comments below. There is a reasonable degree of reasonable overlap between our comments.

I note that you have contacted authors where there were unreplicable results. I think the reader will want a few sentences at the very least explaining the result of these contact attempts (related to the reviewer’s comment on this point) around Line 176 giving the number of articles where contact was attempted and those where at least one author provided information, the number where you received an email but no usable information, and the number who did not reply (including if emails bounced). If you didn’t hear from the corresponding author within a set time (how long was this?), did you also try other authors or check to see that the corresponding author was still actively publishing? Did you look for current affiliation of just that at the time of publication? I wonder if a flow chart might not be a useful supplement to the textual explanation of this also (this could usefully incorporate your statistical test replication and recalculation process).

I think a diagram (also mentioned by the reviewer) would also help show the studies from Khan, the new studies you have added, and the studies excluded from analysis (this is just Rafols?) and where corrected results were used. I’m not sure why the 67 articles/studies (Line 34 in tracked version) becomes 69 (Line 41 in tracked version). Typo? The same number comes up again on Line 245. The data set’s n=67 matches the number used on Lines 34, 171, 209, and 233 which I’m assuming is correct but this would imply Rafols was not excluded?

For the data file, you could add a prominent flag saying “INCLUDED” or “EXCLUDED” (for Rafols only?) to avoid readers getting frustrated when trying to replicate the analyses along with a flag stating when results were “AS REPORTED” or “UPDATED” (as the reviewer notes, the original results would also be useful to include and so having two lines for those studies might be useful to show the original and the values used here—or you could have a separate file with the original results where these were updated).

Line 236 (tracked version) has a misspelling of “statistical”.

Line 239: I’m assuming this should read “not possible“?

Line 248: Perhaps “i.e. a low probability of detecting effect sizes that would be of clinical or other importance if they exist” (related to your clarifications on statistical power earlier in the manuscript)?

Line 263: Perhaps “i.e. the probability of detecting the effect size seen in the study” for a similar reason.

I think the manuscript is much improved and if you can adequately address the reviewer’s and my comments, I think the manuscript will be very close to being publishable. I will echo the reviewer’s point though that the reader must be able to follow the processes you have used and I will again invite you to make use of diagrams, etc. whenever this might make the text easier to follow.

·

Basic reporting

See below

Experimental design

See below

Validity of the findings

See below

Additional comments

Thank you for asking me to re-review this paper concerning Fragility of studies in Sports Medicine. A number of points before my comments: I was not able to access the Raw Data Excel file which produced corrupted text; there appears to be no point by point comments or rebuttal on my earlier referee comments, there is for the Editor and Reviewer 1 and 2. The last observation makes it hard for me to follow thorough how the authors have responded to my comments as you can imagine.

Overall the paper is improved, but I still have significant concerns.
I am very confused by Results Para 1. I think the message might be better shown as a flow diagram (as is done with a study recruitment to analysis flow figure). It is very difficult to follow the authors logic here:
ln 176 What happened when corresponding authors were contacted and responded? Need a detailed breakdown.
Ln 178 are the P values for reg Fischer test the original data or re-worked data? The same comment goes all through this paragraph
Ln184 should the word inconsistent be replaced with incorrect?
How were corresponding authors contacted and how long did they get to respond etc
We desperately need a table of original data and re-worked data, referenced to the original studies.
The authors found new studies to analyse (19 I think Ln 34), but these are not mentioned again outside the abstract, and are not referenced in this paper, and are not discussed in the Results section, nor are commented on in Discussion.
The supplementary files are not referenced in the text, will they be available to the reader? They need to be signposted.
Supplementary file ‘Studies with inconsistencies’ only refers to 3 studies, however Results para 1 suggests another 13 more Ln 177, or 8 Ln 40, explain. Very confusing. Also as in my earlier review it would be useful to have the original author comments seen with the description of the inconsistencies.

---

## Round 0.4 · Minor Revisions

Thank you for your revisions, which I think address all of the outstanding reviewer and editor comments.

There are some copy editing comments that still need to be addressed. I would normally accept the manuscript with just a few of these and leave their correction for the proof stage, but there are quite a few listed below and so I’ll ask you to make these revisions now rather than at the proof stage. I hope you don’t mind addressing these now and I will promise a very quick acceptance once these are all resolved. Some of these are stylistic or subjective (normally indicated by “suggest” or “consider” in my comment) and you are very welcome to not make changes if you prefer for these.

I look forward to seeing the next version of the manuscript and being able to accept that. I think you have written an interesting and important article and one that I anticipate will lead to much discussion. Well done!

Line 23: I’d suggest either the simpler “A recent study concluded…” or “A recently published study concluded…” rather than “A recent study published concluded”.

Line 33: Either add “a” before “small” or change “effect” to “effects”.

Line 35: I suggest that “simulated” rather than “simulated” would be more usual here, or you could say “Three simulations were used…” instead.

Line 48: “A robust study indicates an…” (adding “A” here)

Line 59: You could consider being slightly more precise here and say: “100 identical studies ON AVERAGE only 50…” (adding “on average”) as most times more or fewer than 50 studies will produce significant findings.

Line 63: You could consider replacing “owes to” with “is due to”.

Line 65: “size IS exaggerated compared to THE population value” (adding the words in upper case).

Line 67: You could again consider acknowledging the uncertainty here and say: “…higher ON AVERAGE than…” (adding “on average”).

Line 88: “…it may be THE smallest number of patients…” (adding “the”)

Line 84: “Fragility Index” here could be shortened to “FI” (as defined on Line 79).

Line 92: “Fragility Index” here could be shortened to “FI” (as defined on Line 79 and used on Lines 87 and 88).

Line 94: “were” (plural) rather than “was” (singular) as you have four aims.

Line 108: I’d suggest “resolved” rather than “solved” here. Both are fine but the former is more conventional.

Line 115: “This was A straightforward procedure…” (adding “a”)

Line 128: There is a double period (“. .”) here.

Lines 129 and 130: “Yate’s correction” for each (adding the possessive apostrophe as the correction “belongs” to Yate).

Line 130: “…THE Chi-squared test…” (adding “the”)

Line 131: “…THE first author…” (adding “the”) and “…THE second author…” (same).

Line 141: Should “needed” here either be “provided” or simply deleted as this is the power based on the actual sample sizes?

Line 151: Should “One case…” be “Each case…”?

Line 152: “was similar” suggests that the sample sizes were approximately the same, but I thought they were exactly the same. If this is the case, I’d change “similar” to “the same” here.

Line 153: Change “per one study” to “in each study” or delete “one” here.

Line 154: There seems to be a missing word and perhaps a comma and an “s” in “…the proportion of outcome event randomly generated.”, would could be written as “…the proportion of outcome events, was randomly generated.

Line 157: This is a minor point but presumably negative values were discarded or increased to zero? Or were zero values also not allowed and so values below 1 were increased to 1 or discarded? Presumably the number of events was also rounded to the nearest integer. In any case, these details should be made clear with a brief remark here.

Lines 162: Was this exactly 50% or approximately 50%?

Line 166: “…we identified AN additional…” (adding “an”)

Line 167: I’d suggest just “…comprised 66…” (deleting the existing “of”) or adding “was” before “comprised”.

Line 170: Add “The” or “A” before “Flow chart” here.

Line 172: “response” should be “respond”.

Line 173: “to THE first contact attempt.” (adding “the”).

Line 173: There is a sentence fragment here: “One of the authors and one study group representative.” I’m not quite sure what you mean to say here. Are these the two inconsistency studies in the left-most box in Figure 1?

Line 174: “…THE author…” (adding “the”)

Line 174: “…THE mentioned…” (adding “the”)

Line 174: “Wilcox” should be “Wilcoxon” (https://en.wikipedia.org/wiki/Wilcoxon_signed-rank_test)

Line 174: “…THE group…” (adding “the”)

Line 175: “…THE Chi-squared test…” (adding “the”, although “a” would also work).

Line 175: “…instead OF the…” (adding “of” here).

Line 175: “Fisher’s” (adding “’s”, https://en.wikipedia.org/wiki/Fisher%27s_exact_test).

Line 176: It’s not clear from the text to me where the “three” here comes from. I’m assuming these are the three studies in the second-from-the-left box in Figure 1? One option might be to label the boxes in Figure 1 as “(A)”, “(B)”, “(C)”, and “(D)” and then refer to these letters in the text?

Line 177: Either “A Chi-squared test.” or “THE Chi-squared test.” (so adding either “a” or “the”).

Line 177: Missing space in “of13”.

Line 179: Either “A Chi-squared test.” or “THE Chi-squared test.” (so adding either “a” or “the”).

Line 182: “…the range of p-values TO which a single FI corresponds…” (adding “to”).

Line 186: “…limit of THE 95% CI” (adding “the”).

Line 186: Rather than “prevalent”, I’d suggest “apparent” or “observed”.

Line 191: This says “42% and 43%” but earlier you say “42% and 47%” (Line 38). From Table 1, the latter would appear to be 44%.

Line 193: I wonder if giving two medians might not be confusing, especially since group 1 and group 2 seem somewhat arbitrary to me (apologies if I’m missing something there). Perhaps the overall median of 30 per group (between 12 and 80) would be sufficient to report here?

Line 195: What is the source of these p-values? Mann-Whitney U? This should be mentioned in the statistical methods around Line 121.

Line 199: “with” would be more usual than “to” here, so “…is associated with the p-value…”.

Line 199: Note that there is a spurious comma here: “. ,”.

Line 202: “define A robust study…” (adding “a”).

Line 203: “has” rather than “have” here.

Line 204: “have” rather than “has” here.

Line 210: “with A high” (adding “a” here).

Line 212: Perhaps: “Finally, study validity indicates the strength of the conclusions that can be taken from the study…” (“carried out” would be unusual phrasing here).

Line 216: This is a personal suggestion, but with 66 in the denominator, each unit in the numerator counts for around 1.5% so I would report the percentage as an integer (12%) rather than the more precise 12.1% here.

Line 219: “not explicit” rather than “unexplicit”.

Line 220: I think you mean “not possible” rather than “not impossible” here.

Line 221: I’d say “were due to” rather than “owed to” here.

Line 224: “Another” rather than “Other”.

Line 225: Again, you say “42% and 43%” here whereas you said “42% and 47%” on Line 38.

Line 230: Rather than “to small sample size”, I’d say “with small sample size” (changing “to” to “with”).

Line 234: I’d delete “the” from before “poor” here.

Line 248: If I’m reading this correctly, I’d add “and” before “there should be doubts” here.

Line 252: “…results IN A one-tailed…” (replacing “with” with “in a”).

Line 253: “…that A replication study…” (adding “a” here).

Line 258: I’d delete “is” from the start of this line.

Line 259: “not THE FI” (adding “the” here).

Line 260: Do you mean “…definitive evidence as to whether the finding is indeed true.” Here?

Line 262: “…even if THE prior probability…” (adding “the” here).

Line 263: You could add “for” before “a p-value” here.

Line 264: “In other…” (rather than “In another…”)

Line 265: “…one out OF four studies…” (adding “of”).

Line 269: Do you mean “evaluating” here rather than “estimating”?

Line 270: Do you mean: “Neither does it help to estimate…” here?

Line 281: Technically, if the standard error is known, the sample size is not relevant for comparisons of proportions as the sample size is already included in its calculation (for comparisons of continuous outcomes, the degrees of freedom would still be important for smaller sample sizes when calculating p-values for a given t-test statistic).

Line 282: “finding.” (no “s”) or delete “a single”. Given you then transition into discussing single studies, I suspect deleting “a single” would work better here, but this is entirely up to you.

Line 292: Delete “the” from before “research findings”.

Line 316: Delete “the” before “meta-analyses”.

Line 324: I’d suggest one of “…of small sample sizes BEING utilized…” and “…of THE small sample sizes utilized…” here.

Lines 353–356: I think Colquhoun (2014) should come before Colquhoun (2017) here.

Lines 373–375: Similarly, I think Ioannidis (2005) should come before Ioannidis (2008) here.

Figure 1 seemed a little blurry (for the smaller text) when I zoomed in. Note “Fisher’s exact test” rather than “Fisher exact test” (three instances of “Fisher”) and “Yate’s” (two instances of “Yate”). I appreciate you explaining the meaning of the yellow boxes and wonder if doing the same for the green and salmon coloured boxes would also be possible.

Figure 2: Note missing space in title (“Associationbetween fragility index and p-value”).

Figure 3: Note missing space in title (“Associationbetween post-hoc power and fragility index”). Note also that not all 66 studies came from Khan, et al. and you could use the wording from Figures 2 and 5 for this.

Figure 4: Note missing space in title (“Associationbetween post-hoc power and p-value.”). Same point about Khan, et al.

Figure 5: Note missing space in title (“Associationbetween FI and limits of confidence interval.”).

---

## Round 0.5 · accepted · Accept

Thank you for your revisions. I am very happy to accept your manuscript and look forward to seeing it published and contributing to the literature in this important area.

I will note a few small typos/suggestions that you can easily address at the proof stage.

Line 96: “in the sports medicine and arthroscopic surgery RCTs” would read better to me without “the”.

Line 101: The reference “Khan et al (Khan et al., 2016).” could simply be “Khan, et al. (2016).” to reduce this repetition.

Lines 147–148: I suggest either adding an “s” to the end of “using Mann-Whitney U-test.” as this involves multiple “tests”, or making this “using the Mann-Whitney U-test.”

Lines 155–156: I think you are missing a “, was” in “The number of events, ie. the proportion of outcome events[, was] randomly generated.”

Lines 158–159: You explained this in your response but I think some readers will appreciate also seeing the explanation in the manuscript: e.g. with “The numbers of events in the two groups in each data set were selected randomly from normally distributed values [using the floor of the absolute value to obtain non-negative integers].” or similar here.

Line 168: Perhaps “an additional 330 [candidate] studies”? Or “relevant” or “potential”.

Line 188: You capitalise the “F” in “Figure”, so could do similar for “[T]able 1” here.

Line 197: Might be easier to read with a couple of commas: “Walsh et al.[,] who originally derived the FI[,] suggested”

Line 266: I don’t think you need a comma here in “associated with it, has never been implemented in the field of orthopedics.”

Line 293: I don’t think you need a comma here in “clinical decision making, if we can find a repeat”